# New Insights into Multistep-Phosphorelay (MSP)/Two-Component System (TCS) Regulation: Are Plants and Bacteria That Different?

**DOI:** 10.3390/plants8120590

**Published:** 2019-12-11

**Authors:** Virtudes Mira-Rodado

**Affiliations:** Center for Plant Molecular Biology (ZMBP), Department of Plant Physiology, University of Tübingen, 72076 Tübingen, Germany; virtudes.mira-rodado@zmbp.uni-tuebingen.de

**Keywords:** TCS, MSP, prokaryotes, *Arabidopsis thaliana*, dimerization, protein degradation, phosphorylation and dephosphorylation, phosphatase

## Abstract

The *Arabidopsis* multistep-phosphorelay (MSP) is a signaling mechanism based on a phosphorelay that involves three different types of proteins: Histidine kinases, phosphotransfer proteins, and response regulators. Its bacterial equivalent, the two-component system (TCS), is the most predominant device for signal transduction in prokaryotes. The TCS has been extensively studied and is thus generally well-understood. In contrast, the MSP in plants was first described in 1993. Although great advances have been made, MSP is far from being completely comprehended. Focusing on the model organism *Arabidopsis thaliana*, this review summarized recent studies that have revealed many similarities with bacterial TCSs regarding how TCS/MSP signaling is regulated by protein phosphorylation and dephosphorylation, protein degradation, and dimerization. Thus, comparison with better-understood bacterial systems might be relevant for an improved study of the *Arabidopsis* MSP.

## 1. Review

In order to survive everchanging surroundings, organisms have developed different signaling mechanisms that allow them to sense and respond to environmental cues. The two-component system (TCS) represents a process by which many organisms are able to transduce signals via a phosphorelay that involves a set of proteins. In prokaryotes, these proteins belong to two subfamilies, naming the TCS: The histidine kinases (HK) that auto-phosphorylates a conserved histidine (His; H) residue upon signal perception, and the response regulators (RR) that are activated by accepting the phosphate on a conserved aspartate (Asp; D) residue from the HK [1,2]. Here, RRs are often DNA-binding transcription factors whose affinity to bind to their target promoters is regulated by their phosphorylation state [3,4,5]. When prokaryotic and eukaryotic signaling pathways contain more elements, they are termed multistep-phosphorelays (MSP). In *Arabidopsis* MSPs, HKs contain an additional Asp residue in the so-called receiver domain (RD) and are thus referred to as hybrid HKs. In plant MSPs, a third element, the His-containing phosphotransfer protein (HPt), is also included. Here, HPts act as a mediator connecting HKs and RRs enabling the His-Asp-His-Asp phosphotransfer to take place [6,7]. Unlike plant MSP, in prokaryotic MSPs, the RD and HPt additional modules can appear as single- or as multi-domain proteins. Although the number and character of the proteins involved in a phosphorelay can strongly vary—especially in the bacterial TCS, entailing several HK, HPt, or RRs [8]—the nature of the phosphorelay remains constant and phosphates are transferred from His-to-Asp residues at all times [2,7] (Figure 1). For a more detailed description on bacterial TCS domain architecture and TCS structural basis of signal transduction, I recommend Whitworth’s [9] and Casino’s [10] reviews in Gross and Beier’s book on TCSs in bacteria [11].

The TCS/MSP evolved as a signaling mechanism both in prokaryotes and eukaryotes [12,13]. However, while the TCS is known to regulate many aspects of bacterial life, the identification of MSP’s exact roles in plants remains under heavy study. The model plant *Arabidopsis thaliana* represents the best-understood MSP system in plants. Here, its 11 *Arabidopsis* HKs (AHK) are hybrid HKs containing, in most cases, both an His and an Asp conserved residue within the protein [7,14]. *Arabidopsis* AHKs have been shown to be involved, for instance, in cytokinin (AHK2, 3 and 4) [15,16,17] and ethylene perception and signaling (ETR1 and 2, ERS1 and 2 and EIN4) [18,19,20], act as putative osmosensors in dehydration avoidance and low water-potential responses (AHK1) [21,22,23], female gametophyte development (CKI1) [24], cold stress (AHK2 and 3) [25], freezing tolerance (ETR1 and EIN4) [26], programed cell death (AHK4) [27], and responses to H_2_O_2_ (AHK5) [28,29,30,31]. In the *Arabidopsis* MSP, there are also five canonical *Arabidopsis* HPts (AHP1 to 5) containing a conserved His residue whose function is to transfer the phosphate from the AHKs to the ARRs. In *Arabidopsis*, a pseudo-AHP protein (AHP6), which does not contain the conserved His phosphorylation site, also exists. Finally, the 23 *Arabidopsis* RR (ARR) are divided into three subgroups according to their function and protein structure. All ARRs hold a conserved Asp residue in the RD. In A- (ARR3-9, 15-17) and C-type (ARR22 and 24) ARRs, the output domains are very short, while in B-type (ARR1-2, 10-14, 18-21) ARRs, they contain several structures that are typical for transcription factors: At least one NLS signal and a DNA-binding- and a transactivation domain. Although their direct role as transcriptional activators has only been demonstrated for some of them (ARR1, ARR2, ARR10, ARR11, and ARR18), it is of general consensus that all B-type ARRs function as transcription factors [7,14,32,33,34,35,36,37,38,39]. Similar to the AHPs, *Arabidopsis* holds a family of pseudo-response regulators (PRRs: PRR1-9) that lack the phospho-accepting Asp residue [14]. Within this family, pseudo-response regulators PRR1/TOC1, PRR3, PRR5, PRR7, and PRR9 have been proven to be essential to the function of the circadian clock’s central oscillator [40,41].

Regarding their role in plant development, ARRs are involved in a very wide number of processes, such as the circadian clock (ARR3, 4 and 9) [42,43], lateral root formation (ARR5) [44], responses to light (ARR1 and 12) [45] and cold stress (ARR7 and 1) [25,46], drought and freezing tolerance (ARR5, 7, 15 and 22) [26,47], auxin (ARR7 and 15) [48], responses to ethylene (ARR2) [49], sugar (A-type ARR and PRR7) [40,50] and phytochrome B (ARR4) signaling [51,52], meristem (A-type ARRs) [53], and female gametophyte development (ARR1) [54]. ARRs can also act as phosphatases (ARR22) [38,39,55,56,57]. Ultimately, both A- and B-type ARRs are key factors in cytokinin signaling, with this being one of the best understood roles in *Arabidopsis* MSP [58]. *Arabidopsis* cytokinin signaling starts with the AHK2, AHK3 and AHK4 receptors that auto-phosphorylate in a conserved His-residue upon signal perception and are predominantly located in the endoplasmic reticulum [59,60]. The phosphate is then transferred to an Asp-residue in the RD within the hybrid kinase. Phosphorylated AHKs transfer their phosphate to the cytoplasmic AHPs that then move to the nucleus and subsequently, activate nuclear localized B-type ARRs by a final phosphate transfer. Phosphorylated B-type ARRs act as transcription factors of different cytokinin-regulated genes such as the *A-type ARRs* [7,14,32,33,34,35,36,37,38,39]. The up-regulation of *A-type ARRs* gene expression results in the accumulation of A-type ARR proteins that compete with B-type ARRs for AHP-ARR phosphotransfer; thus, creating a negative feedback loop that reduces the amount of phosphorylated B-type ARRs; consequently, inhibiting cytokinin-regulated gene transcription [39,61,62,63,64]. Similar to A-type ARRs, the pseudo AHP protein AHP6 that lacks the phospho-accepting His residue necessary for His-to-Asp phosphorelay, has been shown to acts as a negative regulator in cytokinin signaling by competing with typical AHPs for the interaction with both AHKs and ARRs [65]. For a more extended description on cytokinin signaling in *Arabidopsis*, I recommend Kieber and Schaller’s recent review [66].

After describing the processes *Arabidopsis* MSP members are engaged in, it is important to note that, at least in processes such as cytokinin signaling, plant MSPs have been shown to involve a very high degree of functional redundancy that occurs at all levels (AHKs, AHPs, and ARRs) of the signaling process. Although bacterial systems also show some degree of redundancy—where one HK might activate several RRs or several HKs recognize a single RR—bacterial TCSs are often linear pathways with all components expressed from a single operon [67,68]. Thus, plant systems might represent a higher degree of difficulty not only in determining which concrete elements define each TCS pathway, but also in understanding how the specificity of these responses are achieved and regulated. 

To control any signaling processes, organisms require efficient means to turn them on and off or even attenuate them. To achieve this, both bacteria and plants have evolved a number of tools. Here, recent publications, describing mechanisms by which bacterial TCSs and plant MSPs regulate signaling in a comparable manner, were reviewed. Such similarities suggest that it might be interesting to consider the better understood bacterial system to recognize new perspectives on how plant MSP signaling is regulated. Here, using the model plant *Arabidopsis*, I focused on three different mechanisms that both bacteria and plants use to finetune TCS/MSP signaling: RR’s protein degradation, protein dephosphorylation, and protein dimerization.

## 2. TCS/MSP Regulation via Response Regulators’ Protein Degradation

As described above, RRs represent the final component in both TCS and MSP signaling. Thus, eliminating such elements would automatically result in the obstruction of the signaling pathway. RR protein degradation has been extensively reported as a mechanism to switch off the TCS signaling cascade in bacteria. For example, the DegS-DegU TCS in *Bacilus subtilis* consists of the DegS HK and its cognate RR DegU [69]. Although the signal that triggers DegS activation has not yet been identified, DegS-DegU phosphorelay results in the activation of DegU that acts as a transcription factor regulating genes involved in many cellular events [70,71]. In addition to the TCS signaling, phosphorylated DegU is a target for the ClpCP protease that is responsible for DegU protein degradation [72]. Therefore, the DegS-DegU TCS pathway represents an example of how different systems can autoregulate signaling, since RR DegU activation by phosphorylation is also coupled to proteolysis by ClpCP. 

In addition to such simple systems, bacteria also include especially complicated mechanisms to modulate signaling by RR’s degradation. In *Caulobacter crescentus*, two MSPs, the CckA-ChpT-CtrA and the CckA-ChpT-CpdR, are composed of the CckA HK, the phosphotransfer protein ChpT, and two RRs (CtrA and CpdR). Both RRs are phosphorylated by the same HK and HPt and are also degraded by ClpXP protease [73,74,75,76,77]. In this system, a tight control of phosphorylation and proteolysis of both RRs adjusts the correct timing of DNA replication during the cell cycle. *Caulobacter crescentus* cell division is asymmetrical, resulting in two types of cells: A motile swarmer cell and a sessile stalked cell. During its cell cycle, the motile swarmer cell differentiates into a stalked cell. This transition coincides with the initiation of DNA replication. During this process, phosphorylated RR CtrA (CtrA~P) prevents DNA transcription by binding to the replication origin [78]. Thus, to allow initiation of DNA replication, CtrA~P needs to be removed from the system [79,80]. In this arrangement, it is interesting that to degrade CtrA~P, the CtrA~P protein needs to form a complex that also includes the unphosphorylated RR CpdR and the ClpXP protease [76,81]. After CtrA~P proteolysis, ClpXP also degrades CpdR to allow cell cycle progression [73]. Thus, chromosome replication in *Caulobacter crescentus* requires the control of ClpXP-regulated proteolysis of two RRs, CtrA and CpdR which, in turn, are members of a common MSP that involves the HK CckA and the HPt ChpT proteins.

The proteolytic machinery that controls the abundance of key regulatory compounds works differently in eukaryotes and prokaryotes. In bacteria, protein degradation typically occurs through ATP-dependent proteases, such as ClpP, HsIV, Lon, and FtsH [82]. These proteins are composed of two domains: The AAA+ domain (ATPases associated with diverse cellular activities superfamily) and the protease domain. These two domains can be integrated into a single protein, as in the Lon and FtsH proteases, or appear as single domain proteins, as it occurs in ClpP and HsIV proteases. For more information on bacterial proteases, see Sauer and Baker’s review [83]. Above, I described the role of two ClpP proteases which, together with Clp ATPases ClpX and ClpC, form the ClpXP and ClpCP proteolytic complexes responsible for protein degradation [84]. ClpP proteases are serine proteases and represent a key role in the regulation of protein turnover in bacteria since, together with Lon, they are estimated to carry out around 80% of cellular proteolysis [85,86]. For further information on bacterial proteolysis, the excellent reviews from Darwin or Moreno-Cinos and colleagues are recommended [87,88].

In eukaryotes, the ubiquitin-proteasome system is the mechanism most widely used to degrade regulatory proteins. In this system, proteins targeted for degradation are first marked by the posttranslational addition of ubiquitin, a process that involves enzymes E1, E2, and E3. Subsequently, ubiquitinated proteins are degraded by the 26S proteasome. Additional information on the ubiquitin-proteasome system can be found in Marshall and Vierstra’s reviews [89,90].

Similar to bacteria, plant systems provide examples where RRs proteolysis results in a mechanism to control MSP signaling. In *Arabidopsis thaliana*, examples both A- and B-type response regulators targeted for proteolysis were found (Figure 1B). 

In the case of B-type ARRs, ARR2 was first reported as an example where RR proteolysis could regulate cytokinin signaling output [91,92]. In these studies, ARR2 degradation was revealed to be mediated by the 26S proteasome, as shown by experiments in *Arabidopsis* protoplasts were ARR2-hemagglutinin (HA) protein levels were measured in the presence or absence of the 26S proteasome inhibitor MG132 [91]. In addition to ARR2, other B-type ARRs, such as ARR1, ARR10, and ARR12, have been also reported as targets of the ubiquitin-proteasome system [93,94,95]. In agreement with this data, Kim and colleagues also reported that different B-type ARRs—ARR1, ARR12, ARR20, and to a lesser extent, ARR2 and ARR10—interact with KISS ME DEADLY (KMD) proteins and that the degradation of ARR1, ARR2, and ARR12 protein is mediated by SCF^KMD^. Thus, the SCF^KMD^ complex negatively regulates cytokinin responses by controlling the levels of a B-type ARR transcription factors [95]. KMD proteins are a family of F-box proteins that, together with Skp1 (S-PHASE KINASE-ASSOCIATED PROTEIN 1), constitute the substrate recognition module of the SCF complex (Skp, Cullin, F-box containing complex), a component of the multi-protein E3 ubiquitin ligase that catalyzes the ubiquitination of proteins bound for proteasome degradation. For a better understanding of the composition and function of SCF ubiquitin ligases and the ubiquitin-proteasome system, Sadanandom and Reitsma’s excellent reviews are recommended [90,96].

These studies also covered the significance of B-type ARR’s proteolysis in cytokinin signaling. First, Kim and colleagues observed that an enhanced B-type ARR protein accumulation altered the expression pattern of *A-type ARRs* when the overexpression of an ARR2 mutant resistant to proteolysis (ARR2^K90G^; substitution of Lys 90 by Gly) resulted in the upregulation of *ARR6* transcript levels [91]. Second, they also showed that ARR2 proteolysis potentially modulates cytokinin sensitivity in multiple developmental traits in *Arabidopsis* and observed that ARR2^K90G^ overexpression affects cytokinin-mediated physiological functions, such as chlorophyll accumulation, hypocotyl and root elongation, callus induction, shoot regeneration, and leaf senescence [91].

It is interesting to note that, as described above in the bacterial system, not all ARRs are susceptible for degradation in the same phosphorylation state. ARR2 degradation requires prior phosphorylation [91], while ARR1 and ARR12 proteolysis occur more rapidly in the non-phosphorylated form [93,95]. As a result, cytokinin signaling, which ultimately results in B-type ARRs phosphorylation, leads to different outputs regarding B-type ARR’s protein turnover. Cytokinin treatment facilitates ARR2 protein degradation, while it promotes ARR1 protein stability [91,94].

Apart from the above described B-type ARRs, A-type ARRs ARR5 and ARR7 have also been shown to be targets of proteasome-mediated degradation [61,97,98,99,100]. The *Arabidopsis* genome encodes several classes of E3 ubiquitin ligases, some of which are represented by large gene families [101]. Li and colleagues reported that one or more members of the CRL-type E3 ubiquitin ligase family (Cullin-RING E3 Ligase) play a key role in A-type ARR proteolysis, and that they require AXR1-dependent RUB (related to ubiquitin protein) modification of their Cullin subunit for optimal activity [97]. CLR E3 ligases comprise a cullin, a RING-box 1 (RBX1, which binds to E2-ubiquitin), and a variable substrate recognition module [101]. Above, I described that in the case of the B-type ARRs, KMD proteins are part of the substrate recognition module. Unlike B-type ARRs, ARR7 does not interact with KMD proteins [95]. Therefore, A-type ARRs that are also degraded through the ubiquitin-proteasome system must use a different mechanism to recognize proteins as targets for degradation.

The ubiquitin-proteasome system does not account for the degradation of all cytokinin signaling components. Thus, in addition to this mechanism, plant genomes encode hundreds of proteases—as in prokaryotes—that are responsible of the regulation of a striking variety of biological processes [102]. In *Arabidopsis* MSP, we can also find such an example: An A-type ARR, ARR4, whose proteolysis is regulated by the ATP-independent serine protease DEG9 (degradation of periplasmic proteins 9) [103]. Although DEG proteases—also called high-temperature requirement A proteins— are found in almost all organisms, including bacteria [104], DEG proteases are not described to participate in the degradation of response regulators in prokaryotes. For example, the response regulators DegU and DegP of *Bacillus subtilis* described above are degraded by ClpCP, but not by DEG proteases [72]. Thus, DEG9 protease seems to have evolved specifically to degrade ARR4 in plants. Nevertheless, similar to the conservation of the cytokinin signaling pathway, orthologs of DEG9 exist in the monocotyledon rice, as well as in the bryophyte *Physcomitrella patens* and the lycophyte *Selaginella moellendorffii*, suggesting that the use of DEG proteases for the cytokinin signaling pathway might be universal in plants [103].

In conclusion, the degradation of both A- and B-type ARRs creates an additional regulation level in *Arabidopsis* cytokinin signaling, since proteolysis of ARRs represents a supplementary mechanism to attenuate the MSP signaling cascade. In *Arabidopsis*, this is achieved by a complex proteolytic machinery in which responses regulators are degraded through diverse mechanisms that are able to distinguish the RR’s phosphorylation state, as well as discriminate between the different ARRs.

## 3. TCS/MSP Regulation via Protein Dephosphorylation

In bacteria, signaling can be switched off by several distinct processes. TCSs and MSPs are signaling mechanisms based on a phosphorelay, which transfers a phosphoryl group from a HK to finally reach and thus activate a RR. Taking this into account, it is conceivable that a system for signal extinction could involve the dephosphorylation of its different components. In bacteria, such a mechanism is observed quite frequently, and the loss of the phosphoryl group can take place through diverse mechanisms. First, HKs may function as phosphatases are thus responsible not only for the phosphorylation, but also for the dephosphorylation of their cognate RRs [105]. Second, RRs can lose their phosphate to their cognate HKs by reverse-phosphorylation [106]. Third, RRs possess auto-dephosphorylation activity that limits the lifetime of their active states [107]. Fourth, bacteria hold a set of auxiliary phosphatases that are not direct members of the TCS family, but are responsible for dephosphorylation of RR [108,109].

The EnvZ-OmpR system in *E. coli*, which monitors osmolarity changes by regulating the expression of outer membrane porins, is a good example on how HKs’ phosphatase activity can regulate a RRs’ function. Here, the activation of the sensor HK EnvZ, by an unknown signal, results in the phosphorylation and, thus, activation of the RR OmpR which, in turn, activates the expression of *ompF* and *ompC* genes that code for outer membrane porins [105,110,111]. The HK EnvZ does not only display kinase, but also phosphatase activity. Consequently, it is able to dephosphorylate the RR OmpR [112]. Because unphosphorylated OmpR does not play a role in porin gene expression [113], by controlling the kinase and phosphatase activities, HK EnvZ could modulate the level of phosphorylated RR OmpR in vivo and ultimately, porin gene expression. On the other hand, additional studies regarding the importance of EnvZ’s phosphatase activity disagree with this hypothesis and postulate that EnvZ’s phosphatase activity actually functions to limit crosstalk between highly homologous TCSs [114]. Further publications question whether HK’s phosphatase activity is at all significant in vivo, basing their arguments on two points. First, experiments to measure HK’s phosphatase activity have ignored in vivo HK to RR ratios. As an example, an in vitro study on EnvZ’s phosphatase activity used a high HK to RR ratio (2:1) [115], while in vivo ratios were reported to be very low (1:35–40) [116]. Second, HKs have a higher affinity to their cognate RRs when they are in their unphosphorylated forms. As an example, EnvZ affinity to unphosphorylated OmpR is three-fold higher than phosphorylated OmpR (OmpR~P) [117]. For further information on the importance of HK’s phosphatase activity in the EnvZ-OmpR system, Kenney’s review is recommended [105]. 

As described above, the dephosphorylation of response regulators can also be achieved through reverse-phosphorylation, which involves transfer of the phosphoryl group from the RR back to the HK. Although this reverse-phosphotransfer leads to the dephosphorylation of the RR, it requires an additional step in which the HK is also dephosphorylated to prevent re-phosphorylation of the RR [118]. Reverse-phosphorilation occurs when the HK to RR phosphotransfer is reverted and the equilibrium in the HK to RR reaction is shifted toward the HK. 

It is important to establish that, although both HK’s phosphatase activity and reverse-phosphorylation mechanisms result in HK-mediated dephosphorylation of the cognate RR, these processes are distinct. Such an affirmation is based on the fact that HK’s phosphatase activity does not require HK’s conserved histidine residue responsible for HK to RR phosphotransfer, indicating that dephosphorylation of the RR~P by HK’s phosphatase activity does not involve a reverse-phosphotransfer mechanism [119,120,121,122].

Reverse-phosphorylation has been reported in several two-component regulatory systems, including NRII-NRI [118], CheA-CheY [106], and ArcB-ArcA [123] in *Escherichia coli* or PhoR-PhoP in *Bacillus subtilis* [124]. 

The *Escherichia coli* CheA-CheY chemotaxis pathway is a well-studied TCS that can be used as an example for reverse-phosphorylation. Chemotaxis in bacteria refers to motility changes triggered by sensing of chemical gradients. The molecular machinery which leads to bacterial movement is composed of many components. Here, I mainly focused on the TCS elements: HK CheA and its cognate RR CheY. In this TCS, specialized chemoreceptors termed methyl-accepting chemotaxis proteins (MCPs) are assembled as multimeric complexes, sense environmental signals and, through a scaffolding protein, transmit the signal to the HK CheA. This results in the autophosphorylation of the HK and, subsequently, the phosphorylation of two RRs: CheY and CheB. Here, CheY directs the motility of the flagellar motor, while CheB is a methylesterase that regulates the methylation state of the MCP receptors. For a more detailed description of the *Escherichia coli* chemotaxis pathway, Backer’s review is recommended [125].

Reverse-phosphorylation in the CheA-CheY TCS was described when Steward and colleagues observed that radioactive labelled phosphate (^32^P) was partially transferred from the phosphorylated RR CheY~^32^P to the HK CheA [106]. These studies also reported that high concentrations of unphosphorylated CheA affect the equilibrium of the CheA~P + CheY ↔ CheA + CheY~P reaction. Here, the rate of HK to RR phosphotransfer was inhibited by increasing the concentration of unphosphorylated CheA [106].

If we concentrate on the RR’s side, it should be pointed out that these proteins possess an intrinsic auto-dephosphorylation activity that controls the lifetime of their active state by limiting the duration of their phosphorylated state. Here, RRs have been reported to spontaneously lose their phosphate at different rates. As an example, the half-life of a phosphorylated RR in bacteria can range from seconds in the case of CheY and CheB [126,127] to several hours in the case of OmpR and Spo0F [110,128]. To distinguish between the different processes, it is worth mentioning that response regulator’s auto-dephosphorylation activity requires an active Mg^2+^ site to mediate catalysis [129], which is aided by conserved threonine/serine [130] and lysine [131] residues [109]. 

In addition to the RR’s auto-phosphatase activity, RR’s dephosphorylation rate can be catalyzed by auxiliary phosphatases that increase the RR’s auto-dephosphorylation rate. In bacterial systems, four different classes of RR phosphatases have been identified: CheZ, CheC/CheX/FliY, SpoOE, and Rap families [108,109,132]. An example is the well-characterized dephosphorylation of the chemotaxis RR CheY that can be achieved by one or more of these phosphatases (CheC, CheX, and/or CheZ). For example, CheZ enhances the rate of CheY auto-dephosphorylation by a factor of about 100 [133]. The CheZ and CheC/CheX/FliY families, despite different overall structures, employ identical catalytic strategies for the hydrolysis of the phosphoryl group. The exact mechanism used by the Rap phosphatases to catalyze the dephosphorylation of RRs has not yet been elucidated. In the case of the Spo0E phosphatase family, its members contain sequence and structural features that suggest they could use a similar catalytic strategy to that of the chemotaxis phosphatases [109]. For a more detailed description on the phosphatases regulating chemotaxis response regulator CheY, Silversmith and Wolanin’s reviews are recommended [107,109]. At this point, Ninfa’s chapter on phosphatase activity of TCS proteins [132] in Gross and Beier’s book on TCSs in bacteria should also be mentioned [11].

Protein dephosphorylation is also a system that plant MSPs use to regulate signaling. In *Arabidopsis*, like in bacterial TCSs, some HKs have been proven to possess phosphatase activity (Figure 1B). In cytokinin signaling, receptor kinases AHK2, 3, and 4 are the first components to be phosphorylated in the MSP. Here, a phosphate is transferred from the AHKs to the AHPs to finally reach the ARRs and trigger cytokinin-mediated gene expression. It should be stressed that, while AHK2 and 3 can only function as kinases, AHK4 holds a dual role in cytokinin signaling: It acts as a kinase in the presence of the hormone, and it can also function as a phosphatase that de-phosphorylates AHPs in the absence of cytokinin [134]. In these experiments, Mahonen and colleagues examined the kinase and phosphatase activities of wild-type AHK4 and the *wooden leg* (*wol*) allele of AHK4 that carries a T278I amino acid change (AHK4^T278I^) in the putative extracellular cytokinin binding region and lacks cytokinin binding activity [16]. In the presence of cytokinin and P^32^-labelled ATP, wild-type AHK4 was able transfer P^32^ to AHP1, AHP2, AHP3, and AHP5 proteins, while the AHK4^T278I^ mutant showed only traces of kinase activity [134]. To test phosphatase activity, AHP1, AHP2, AHP3, and AHP5 carrying P^32^-phosphoryl groups were incubated with either wild-type or T278I-mutated AHK4. The radioactivity of each AHP decreased, and both AHK4 and AHK4^T278I^ were transiently phosphorylated [134]. It should be pointed out that phosphatase activity did not appear to be affected by cytokinin treatment, but required the conserved phospho-accepting aspartate (Asp973), since the mutation of Asp973 to Asn (AHK4^D973N^) completely abolished AHK4 phosphatase activity [134]. In conclusion, the transient phosphorylation of AHK4 suggest that this is an example of reverse-phosphorylation, and that the phosphate group flows from the AHP to the conserved Asp973 of the AHK4 protein and is finally released as inorganic phosphate [134]. Similar to AHK4, CKI1, a non-cytokinin receptor belonging to the AHK family, holds phosphatase/reverse-phosphorylation activity toward AHP1 and AHP2 in vitro [135]. 

Regarding the ARRs in *Arabidopsis*, several in vitro experiments have been carried out to elucidate different AHP-ARR phosphorelays. For example, AHP1 and AHP2 have been reported to be able to transfer their phosphate groups to A-type ARR3 and ARR4 [51,136,137]. The B-type ARR11 has also been shown to gain a phosphate group when incubated with AHP2 [34]. Although these experiments did not specifically analyze ARR to AHP reverse-phosphorylation or ARR auto-dephosphorylation in A- and B-type ARRs, they provide insight into the lifetime of ARR phosphorylation. ARR4 remained phosphorylated for only five minutes [136], while ARR3 and ARR11 continued to be modified for up to 10 minutes [34,136], suggesting that not all ARRs have equal auto-dephosphorylation rates (Figure 1B).

In comparison to the above described A- and B-type ARRs, C-type ARR22 represents a peculiar case in the *Arabidopsis* AHP-ARR phosphorelay. When radioactively phosphorylated AHP5 (AHP5~P^32^) was incubated in vitro with ARR22 protein, the radioactive phosphoryl group on AHP5 disappeared within one minute, but ARR22 was not phosphorylated itself [57]. In addition, Kiba and colleagues showed that ARR22 catalytic ability requires the conserved Asp74 residue involved in the MSP, since the mutation in ARR22’s Asp74 residue to Asn resulted in AHP5 staying phosphorylated for a prolonged period of time [57]. These events were best interpreted by assuming that the phosphoryl group on AHP5 was transiently transferred onto ARR22 and then rapidly released from ARR22 [57]. This, suggests that ARR22’s auto-dephosphorylation rate is more rapid than that observed for A-types ARR3 and ARR4 [51,136], the B-type ARR11 [34], or AHPs [34,51,136]. As a result of ARR22’s high auto-dephosphorylation rate, ARR22 has been termed to function as a phosphohistidine phosphatase [38,39,55,56,57] (Figure 1B). 

Compared to bacterial TCSs, little has been done do elucidate how *Arabidopsis* AHKs and ARRs regulate their own dephosphorylation or that of other MSP components. Thus, it remains unknown whether AHK4 and ARR22 are the only elements in *Arabidopsis* MSPs that can exert phosphatase/reverse-phosphorylation activity. In addition, there have been no auxiliary phosphatases described in plants similar to those present in bacteria (Figure 1B). Therefore, the question remains whether phosphatases involved in modifying ARRs’ dephosphorylation rates exist in plant MSPs.

## 4. TCS/MSP Regulation via Protein Dimerization

In bacterial TCSs, histidine kinases are multi-domain proteins. Although they can show a considerable amount of architectural variety, the great majority of HKs contain two structures that are highly conserved: A sensory domain and a kinase domain. The sensory domain is usually a transmembrane domain with an extracellular extension. Here, as an example, EnvZ’s sensory portion is responsible for monitoring osmolarity changes in *E. coli* [138]. It should be noted that although such domains are very common in bacterial HKs, proteins, such as the HK CheA from the *Escherichia coli* chemotaxis pathway, lack any type of sensory domain [139]. Regarding the kinase domain, this structure usually contains a two-helix bundle (DHp) that allows dimerization, a catalytic domain that binds ATP, and a histidine phosphotransfer domain containing the His residue responsible for autophosphorylation and phosphotransfer [140,141,142]. In some bacterial HKs, a RD typical of RRs is fused directly to the HK, creating a hybrid kinase similar to those found in plants where HKs contain both conserved His and Asp residues required for MSP. 

Although some bacterial HKs, like those belonging to Class III, lack a dimerization domain [143], most bacterial HK proteins actually contain such a structure. Thus, HK dimerization is normally required for proper functioning and can be used as a tool to finetune TCS signaling. For more information on HKs domain’s architecture, Kenney’s and Bhate’s reviews are recommended [105,144].

When HKs dimerize through the two DHp helices, they form a four-helix bundle that contains the conserved His residue responsible for the phosphorelay and the globular catalytic ATP-binding domains, protrude on either side of the dimer helical stem [142]. Here, autophosphorylation of the conserved His residue has been shown to occur both in *cis* and in *trans*, and it is determined by the handedness of the loop between the two DHp helices [145]. In examples such as HKs CheA or EnvZ, where the loop turns right, the catalytic domain of one monomer is closer to the histidine of the dimeric partner, and autophosphorylation proceeds in *trans* [141,146]. In cases like the HKs ArcA or PhoR, where the loop turns left, the catalytic domain is closer to its own histidine and autophosphorylation occurs in *cis* [144,145,147,148]. 

Like bacterial HKs, plant AHKs have also been shown to dimerize. Within the cytokinin receptors (AHK2, 3 and 4), co-affinity purification assays revealed that the cytoplasmic domains of AHK2 interacted with each other [149]. In addition to the homodimeric interaction of AHK2, AHK3’s cytoplasmic region could interact with AHK2 and AHK4 domains [149]. The cytoplasmic region in these proteins includes both the His-containing, phospho-accepting region and the ATP-binding catalytic domain [149]. Further studies carried out in vivo not only corroborated AHK3-AHK4 heterodimerization, but also added that AHK3 could form homodimers [59].

Extensive studies regarding ethylene receptors abilities to oligomerize have also been performed on *Arabidopsis*. Here, ETR1 has been shown to dimerize [150] and form heterodimers with all the other ethylene receptors (ERS1, ETR2, ERS2, and EIN4) in pulldown experiments [151] (Figure 1B). Membrane recruitment assays (MeRA), a novel approach to study protein–protein interactions in planta, revealed that all *Arabidopsis* ethylene receptors formed homo- and heteromeric complexes in all possible combinations in the endoplasmic reticulum of living plant cells [20]. This data was also supported by interaction analysis using the mating-based split-ubiquitin system [20,152].

Regarding the dimerization domain of AHKs, information is currently only available on the high-resolution structure solved for the dimerization and histidine phosphotransfer (DHp) domain of ERS1 [59,150,151,153]. The DHp domain of ETR1 overall resembles that of bacterial HKs, since dimerization is also achieved by the interaction of each monomer through a hairpin portion that forms a four-helix bundle structure in the dimer [153]. Based on the loop topology, Pekarova and colleagues proposed that ERS1 autophosphorylation could occur in *trans* similar to EnzV or CheA from *E. coli* [141,146,154]. 

The possibility of *trans*-phosphorylation in AHK dimers raises the question of whether receptor’s heterodimerization could be used as a cooperation system between different AHKs. Grefen and colleagues have proposed such a mechanism for the ethylene receptors where they suggested that, upon autophosphorylation on the histidine residue, ETR1 or ERS1 receptors could transfer their phosphate to the aspartate within the RD of receptors like ETR2, ERS2, or EIN4 that lack the necessary residues for histidine kinase activity [20]. 

Similarly, Gao and colleagues [151] suggest that the ethylene receptors may exist as multimeric clusters in a manner potentially analogous to that found in the histidine kinase-linked chemoreceptors of bacteria [125]. Bacterial chemoreceptors form higher-order clusters to respond to ligand binding in a coordinated way [125]. Hence, Gao and colleagues have proposed that, similar to bacterial chemoreceptors, interactions between ethylene receptors may be used as a mechanism to fine tune the signal output [151].

In bacterial systems, RRs are composed of a RD that contains the conserved phosphor-accepting aspartate residue [3] and an output domain that can vary enormously [155]. Nearly 50% of RRs, including OmpR, PhoB, RcsB, or ArcA, function as transcription factors that contain a DNA-binding domain as output domain [5,156]. RR’s receiver domains typically contain a (αβ)_5_ structure that alternates β-strands with α-helices to form a five-stranded parallel β-sheet surrounded by two α-helices on one side and three on the other [3]. RDs dimerize through their (αβ)_5_ structure by four different structural arrangements that differ on the α-helix and β-strand numbers involved in packing [5]. RR’s dimerization seems to play a crucial role in their regulation, since the largest differences between RR’s phosphorylated (active) and non-phosphorylated (inactive) conformations typically occur on the α4β5α5 face of the RD [3,5,157]. Thus, it makes sense that the activation of RRs that act as transcription factors typically involves dimerization or higher-order oligomerization [4,158,159,160,161,162,163,164,165,166,167,168].

The *E. coli* PhoR-PhoB TCS is a good example of how dimerization affects the activity of the RR transcription factor PhoB. Under inorganic phosphate starvation conditions, the transmembrane sensor kinase PhoR activates PhoB [169], which then induces gene expression of the phosphate (pho) regulon, whose products are involved in phosphorus uptake and metabolism [170].

Structural analyses of PhoB have elucidated that the RR dimerizes as either active or inactive homodimers [168,171]. These distinct conformations are mediated by PhoB’s receiver domain: In the active (phosphorylated) conformation, dimerization is promoted via the α4β5α5 face, while in the inactive (non-phosphorylated) state, α1α5 dimers are observed [168]. Studies by Mack and colleagues basically support a model of PhoB regulation, where phosphorylation of the RR results in its dimerization following a specific conformation that enhances DNA-binding and, consequently, gene transcription regulation [168]. 

This, of course, begs the question of whether plant RRs, which also act as transcriptional activators, require both phosphorylation and dimerization to be able to bind DNA and activate gene transcription. 

In *Arabidopsis*, B-type ARRs act as transcriptional activators. Some B-type ARRs, like ARR14 and ARR18, have been shown to form homodimers [36,149] (Figure 1B). Similar to that observed for bacterial RR PhoB, ARR18’s dimerization ability is regulated by its phosphorylation state [36]. It has already been described that mutations of the conserved Asp residue in plant RRs (responsible for MSP) to either a Glu (ARR^DE^) or an Asn (ARR^DN^) results in RRs that mimic a phosphorylated or a non-phosphorylated state, respectively [33,36,38,39,49,51]. In planta FRET-FLIM, when ARR18’s conserved Asp70 was mutated either to a glutamate (ARR18^D70E^) or to an asparagine (ARR18^D70N^), dimerization of the ARR18 protein was only observed when both monomers were in the same phosphorylation state [36]. In addition, reporter gene activation assays also established that ARR18 requires activation by phosphorylation to function as a transcription factor, since only wild-type ARR18 and ARR18^D70E^ were able to trigger gene activation [36]. These experiments open the possibility that ARR18 forms active (phosphorylated) and inactive (non-phosphorylated) dimers similar to bacterial RRs.

Regarding the structural components implicated in plant RR’s dimerization, to date, there is not much information available on which domains could be involved. In bacterial systems, receiver domains present in both hybrid HKs and RRs are conserved structures composed by an (αβ)_5_ domain [3,156,172]. The crystal structures of plant HKs ETR1, CKI1, and AHK5 receiver domains have been solved and also show a high level of conformational conservation to prokaryotic RDs containing the (αβ)_5_ core [154,173,174,175]. In conclusion, although there seems to be a general structural conservation regarding receiver domains within prokaryotic and eukaryotic kingdoms, no ARR’s receiver domain structure has yet been solved. Thus, it remains open whether in plant systems, RR’s dimerization is promoted via a α4β5α5 interface similar to bacteria.

Similar to the situation in the majority of prokaryotic RRs, plant B-type ARRs that act as transcription factors combine their RD with an output domain containing a DNA-binding motif [4]. In plants, the structure of the DNA-binding motif of *Arabidopsis* transcription factor ARR10 has been resolved [176]. Similar to bacterial RRs like PhoB, it is composed by a helix-turn-helix (HTH) motif that includes three α-helices (α1, α2 and α3) and type I and II β-turns, thus representing a common DNA-recognition unit present in numerous DNA-binding proteins [154,176,177,178]. 

To summarize, although there seems to be a common thread on how bacterial and plant RRs regulate their transcriptional activity, there is still not enough information in the plant kingdom to determine whether *Arabidopsis* ARRs also require phosphorylation and dimerization to be able to bind DNA and activate gene transcription.

Further oligomerization studies in *Arabidopsis* MSP elements established that hetero-dimerization also seems to take place within B-type ARRs [149]. Using the yeast-two-hybrid system, Dortay and colleagues concluded that ARR14 also interacts with ARR2, opening the possibility of both ARRs to form hetero-dimers [149] (Figure 1B). However, the functional relevance of plant RR’s hetero-dimerization has not been elucidated yet. Remarkably, bacterial systems seem to use the mechanism of homo- and heterodimerization to alter DNA binding and target gene specificity. For example, in the *Erwinia amylovora* RcsC-RcsD-RcsB MSP (HK RcsC, HPt RcsD and RR RcsB), the RR RcsB can bind DNA either as a homodimer for transcription activation, or as a hetero-dimer with RcsA, a response regulator-like protein, to regulate a distinct set of genes [179].

Regarding the C-type ARRs (ARR22 and ARR24), no interaction studies have been carried out to date to determine whether these proteins are able to dimerize (Figure 1B). Within the A-type ARRs, ARR5 has been show to homodimerize both in vitro using coimmunoprecipitation assays and in vivo by luciferase complementation imaging assays [180] (Figure 1B). Finally, to complete the MSP, we also have some information regarding interaction studies on the AHP proteins. Here, we find some discrepancies: Experiments using the yeast-two-hybrid system concluded that AHPs do not form either homo- or heterodimers with each other [149], while steady-state fluorescence polarization studies on the ETR1-AHP1 complex formation indicated that AHP1 forms a homodimer [181]. Bimolecular fluorescence complementation studies of AHP2 also showed that this protein can homodimerize [182] (Figure 1B).

In conclusion, analogous to bacterial systems, recent publications have reported that in *Arabidopsis*, dimerization takes place in histidine kinases, phosphotransfer proteins and response regulators [20,36,59,149,150,151,180,181,182]. In bacterial systems, phosphorylation-dependent dimerization is a widely spread mechanism for response regulators to bind their cognate DNA sequence and activate target genes [5]. Whether or not plant B-type ARRs exert their function as transcription factors by similar means has not yet been determined. Altogether, a deeper understanding on how MSP elements oligomerize will allow us to determine the functional importance of homo- and heterodimerization in plant MSP signaling regulation.

In summary, the goal of this review was to compare the well-characterized bacterial TCS with its equivalent MSP in *Arabidopsis*. Such an approach has already been proven interesting in Schaller’s and Horak’s reviews [183,184]. Here, Schaller and colleagues described the evolution from bacterial TCS to plant MSP in regulatory processes such as cytokinin and ethylene hormone signaling or phytochrome-mediated perception of light [183]. Horak and colleagues discussed different mechanisms that could create specificity in MSPs, including protein recognition and the roles of kinase and phosphatase activities and their kinetics as well as MSP elements´ specific expression patterns [184]. 

In this review, three main mechanisms that both bacteria and plants use or may use to finetune TCS/MSP signaling were described: RR’s protein degradation, protein dephosphorylation, and protein dimerization. Due to the similarities observed between eukaryotes and prokaryotes in these regulatory processes, I believe that the better understood bacterial TCS could be used as a crucial reference system to study plant MSPs in more detail. Within this regard, structural-function studies in bacteria should illuminate conserved mechanisms with those TCS components preserved in plants. To date, most work has focused on the “big picture” of the MSP in plants, including which components are redundant, what type of crosstalk is present with other signaling pathways, and how the MSP generally works. In contrast, the bacterial literature is laden with crosstalk studies between specific TCS members and their interaction at the level of phosphotransfer signal transduction. This is important, as it is this level where the TCS can integrate and convey information from and to other pathways.

## Figures and Tables

**Figure 1 plants-08-00590-f001:**
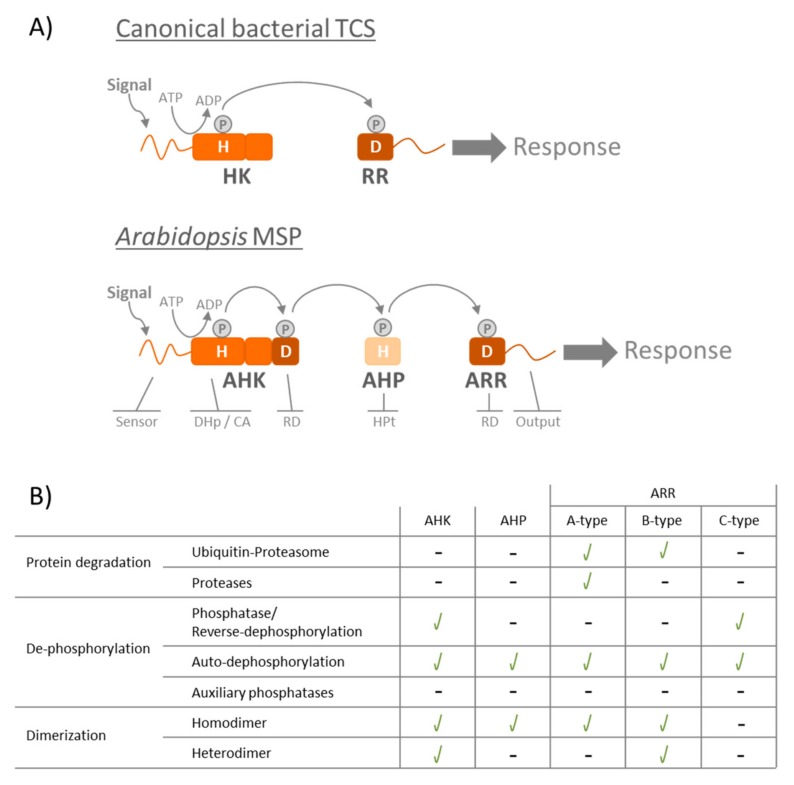
(**A**) Diagram of a canonical bacterial two-component system (TCS) and a multistep-phosphorelay (MSP) in *Arabidopsis*. Upon signal perception, the histidine kinase (HK and AHK) auto-phosphorylates on a conserved His residue (H). The phosphorelay is carried out from His to Asp (H to D) to finally activate the response regulator (RR and ARR) that triggers the response. Canonical bacterial TCS only include two proteins: AHK and a RR, while Arabidopsis MSP also include His-containing phosphotransfer proteins (AHPs) that connect the phosphorelay between the AHK and the ARR. The different TCS/MSP proteins contain the following domains: Sensor: Sensor domain; DHp: Dimerization domain; CA: Catalytic domain; HPt: Histidine-containing Phosphotransfer protein; RD: Receiver Domain; Output: Output domain. (**B**) Summary of different processes that regulate TCS/MSP signaling. All the mechanisms depicted in this table represent different regulatory mechanisms that bacteria use to modulate TCS signaling. In this table, we describe whether these processes also occur in *Arabidopsis* MSP. (Green check mark): Mechanism shown to occur in Arabidopsis MSP. (-): Process that either does not take place in that specific type of protein or where there is still no research available in *Arabidopsis* literature.

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
