# Peer review of "New Insights into Multistep-Phosphorelay (MSP)/Two-Component System (TCS) Regulation: Are Plants and Bacteria That Different?"

_plants, 2019, doi:10.3390/plants8120590_

Round 1
Reviewer 1 Report
The manuscript reviews a unique signaling strategy of plants and bacteria. Owing to the importance of two component systems in important signaling processes such as cytokinin signaling for example, this review is timely and necessary. My only serious concern which prohibits immediate acceptance is that unlike what the abstract implies, the reference list is not up-to-date as it should be and less than 10% of the listed refs come from the past 5 years. Therefor in order to accept the review, I suggest a comprehensive updating of the cited refs and to reform the text accordingly.
Author Response
I would like to thank the reviewer for the comments and helpful criticisms.
I have carefully tried to address the reviewer's points. The reviewer's original comments are written in blue and my response to each point in black to facilitate the reading.
Reviewer 1:
The manuscript reviews a unique signaling strategy of plants and bacteria. Owing to the importance of two component systems in important signaling processes such as cytokinin signaling for example, this review is timely and necessary. My only serious concern which prohibits immediate acceptance is that unlike what the abstract implies, the reference list is not up-to-date as it should be and less than 10% of the listed refs come from the past 5 years. Therefor in order to accept the review, I suggest a comprehensive updating of the cited refs and to reform the text accordingly.
In my review, I compare the two-component system (TCS) in bacteria and plants and describe how these two organisms use similar mechanisms to regulate TCS signaling via processes such as protein phosphorylation and de-phosphorylation, protein degradation and dimerization. As I mention in the abstract, bacterial TCS “has been extensively studied and thus, it is generally well understood”. Accordingly, the literature regarding bacterial TCS has been available for many years.
Opposite to bacterial TCS, the mechanisms by which plants regulate TCS signaling are only now starting to be described and, thus, as the reviewer points out, they only represent a small percentage of the review´s references.
My intention in this review is to point out that both bacterial and plants TCSs share similar mechanisms to regulate signaling and that the comparison with the better understood bacterial system might be relevant for an improved study of plant TCS. To do so, I cited the last findings in plant TCS regulation (focusing on Arabidopsis) but also the extensive literature from their bacterial counterparts that has been available for very long.
In my experience, the plant TCS field rarely recalls bacterial TCS findings and, therefore, direct comparisons between both signaling mechanisms are rare. I hope that reviews like mine promote such associations, because they are of great interest.
Reviewer 2 Report
The review submitted by Mira-Rodado summarizes the current knowledge on Arabidopsis Two Component System (TCS), giving emphasis to the mechanisms by which this signaling transduction system is regulated. In particular, the author highlights the different modes of TCS regulation by combining and comparing previous and recent information available in Arabidopsis and bacteria, underlining differences and similarities.
Overall, it is an interesting and a well-constructed review which provides to those working on the topic and to the less familiar readers a brief and updated overview. I have only some minor comments and suggestions to the author which are given below:
-For comparison of TCS aspects between organisms, I would follow the way it is written in the title i.e. “plants and bacteria” rather than “prokaryotes and eukaryotes”. The use of the latter discrimination can be easily misunderstood (e.g. in line 458).
-Along the same lines, the author might consider specifying in the introduction the prokaryotic and eukaryotic groups in which TCS are present.
-Regarding Figure 1: The addition of a, similar to the existing one, table summarizing the aspects of TCS regulation in bacteria would definitely be of help. Besides, a more comprehensive legend is warranted.
-Line 482, the possibility of oligomerization of ARR5 is not excluded in ref. 182. Is the word “also” needed in that sentence?
-The author might combine the last two paragraphs of the manuscript in a separate conclusion section.
- Regarding the evolution of TCS, the author recommends already relevant reviews. However, the addition of a few words with respect to the presence of two component signal transduction systems in chloroplasts (and mitochondria) could be useful. Besides, from an evolutionary perspective, it might also be mentioned somewhere that certain responses to comparable stimuli in bacteria and in plants involve the activation of TCS and MSP, respectively (e.g. temperature alterations, changes in the redox status).
- When mentioning names of the authors in the text, please replace throughout: “and colleges” with: “and colleagues” or “et al.”
-The abbreviations list needs to be updated.
-Line 161, replace with “these” data
-Line 253, replace “phosphorilation” with: “phosphorylation”
-Line 348, add “to” instead of “do”
-Line 403, replace “hereto-..” with “hetero-..”
-Line 439, “act transcriptional activators” add “as”
Author Response
Please see the attachement

Reviewer 3 Report
Dear Author
Your review paper is of first importance for people working on multistep phosphorelay system in plants. The review is well documented and very well written. The comparison between prokaryotes and eukaryotes is really important for scientists to better understand MSP in plants and to find new fields of investigation. Congratulation for this huge work.
I have just some comments listed below
Abstract : line 8 and 9
I have a problem when you write TCS in line 8, and line 9 you explain that the TCS is composed by three partners. In my opinion, I think better for the readers to avoid misunderstanding to reserve TCS for prokaryots.
Review
line 30: I think you are describing prokaryots RR, and if I am right, I have a probel with the references ciet (3-7) because the RR are from plant (since you wrote line 25 in prokaryotes..).
Line 33 and in text: You wrote HP for Histidine containing phsophotransfer protein but in the Figure 1 A, it is writen HPt. In my opinion, HPt is more precise but if you choose HP, you have to change it in the Figure.
In the table, about the dimerization of HPt proteins, there is one article which mention the dimerization of these proteins, Punwani et al., 2010. May be you can include this article and change in the table because the homodimerization has been shown, and change in the text line 486-487.
I Think it is important to change TCS line 103 by MSP
Change Figure 1 by Figure 1B in different parts of the text (line 155, 306, 335, 347, 352, 390, 440, 474, 481, 488)
The same, you have to change colleges by colleagues (line 275, 311, 340, 400, 404, 408, 411, 432) and Phosphorylation line 253, do by to in line 348, "a" by "an" line 359, and "were" by "where" line 404. Also in abbreviations, add Histidine for HPt protein.
Round 2
Reviewer 1 Report
The author provided adequate answer